# Effects of Post-Exertional Malaise on Markers of Arterial Stiffness in Individuals with Myalgic Encephalomyelitis/Chronic Fatigue Syndrome

**DOI:** 10.3390/ijerph18052366

**Published:** 2021-02-28

**Authors:** Joshua Bond, Tessa Nielsen, Lynette Hodges

**Affiliations:** School of Sport, Exercise and Nutrition, Massey University, Palmerston North 4442, New Zealand; enzed007@live.com (J.B.); tessa.nielsen94@gmail.com (T.N.)

**Keywords:** myalgic encephalomyelitis, chronic fatigue syndrome, arterial stiffness, post-exertional malaise

## Abstract

*Background:* Evidence is emerging that individuals with myalgic encephalomyelitis/chronic fatigue syndrome (ME/CFS) may suffer from chronic vascular dysfunction as a result of illness-related oxidative stress and vascular inflammation. The study aimed to examine the impact of maximal-intensity aerobic exercise on vascular function 48 and 72 h into recovery. *Methods:* ME/CFS (*n* = 11) with gender and age-matched controls (*n* = 11) were randomly assigned to either a 48 h or 72 h protocol. Each participant had measures of brachial blood pressure, augmentation index (AIx75, standardized to 75 bpm) and carotid-radial pulse wave velocity (crPWV) taken. This was followed by a maximal incremental cycle exercise test. Resting measures were repeated 48 or 72 h later (depending on group allocation). *Results:* No significant differences were found when ME/CFS were directly compared to controls at baseline. During recovery, the 48 h control group experienced a significant 7.2% reduction in AIx75 from baseline measures (*p* < 0.05), while the matched ME/CFS experienced no change in AIx75. The 72 h ME/CFS group experienced a non-significant increase of 1.4% from baseline measures. The 48 h and 72 h ME/CFS groups both experienced non-significant improvements in crPWV (0.56 ms^−1^ and 1.55 ms^−1^, respectively). *Conclusions:* The findings suggest that those with ME/CFS may not experience exercise-induced vasodilation due to chronic vascular damage, which may be a contributor to the onset of post-exertional malaise (PEM).

## 1. Introduction

Myalgic encephalomyelitis/chronic fatigue syndrome (ME/CFS) is a debilitating illness, affecting tens of millions of people globally [1]. Within its population of 4.5 million, New Zealand has an estimated 20,000–25,000 ME/CFS patients [2]. ME/CFS has no universally accepted etiology, case definition or diagnostic criteria. At its foundation, ME/CFS is characterized by experiencing persistent and relapsing fatigue for longer than 6 months, which is not relieved by rest. This fatigue is worsened by bouts of physical or mental exertion and is referred to as post-exertional malaise (PEM). A lack of an established pathophysiological pathway for PEM further adds to the confusion and complexity surrounding this illness [3].

Cardiovascular abnormalities have been reported across various cohorts of ME/CFS in research, with reports of impaired blood pressure and heart rate regulation, autonomic dysfunction, and irregular heart conduction [3,4,5]. Accumulating evidence suggests that chronic oxidative stress may be responsible for symptoms observed in ME/CFS populations, consequently damaging arterial endothelial function [5,6]. Abnormal oxidative stress in ME/CFS is thought to be caused by increased lipid peroxidation and excessive presence of free radicals [5]. These observations occurred independently of other cardiovascular risk factors such as hypertension or obesity, implying that CFS patients may exist in a chronic pro-oxidant state as a result of their illness [5]. A series of studies also found differentially expressed genes in ME/CFS patients with evidence of immune modulation, oxidative stress, cell apoptosis, and B cell dysfunction [7,8] further implying the occurrence of oxidative damage on a cellular level in ME/CFS.

Chronic oxidative stress in ME/CFS patients has led to the notion that ME/CFS is also a pro-inflammatory illness [6], as chronic oxidative stress is a cause and trigger of vascular inflammation [9]. This concept is supported by findings that indicated increased production of inflammatory markers in ME/CFS such as C-reactive protein (CRP), cyclo-oxygenase-2, and inducible nitric oxide synthase [10,11]. Such markers of chronic inflammation have been strongly associated with impairing arterial properties in even apparently healthy populations and is associated with increased cardiovascular risk [12,13]. The relationship between arterial stiffness and chronic inflammation has also been established through increased measures of augmentation pressure (AP), augmentation index (AIx75), and pulse wave velocity (PWV) in other populations including type II diabetics [14], persons aged over 55 [13] and even in healthy individuals [12].

Arterial stiffness is a general term used to communally describe three main characteristics: compliance, distensibility and elastic modulus, in relation to the arterial vascular system. These arterial properties differ amongst location of the arterial tree, as blood vessels differ in size, muscle, and collagen/elastin content [15]. Increased measures of arterial stiffness have demonstrated to be a strong predictor for increased stroke, heart failure, atherosclerosis, myocardial infarction, and coronary artery disease, independent of blood pressure [16]. PWV is a systemic, non-invasive method of assessing arterial stiffness by determining the speed in which a pressure wave moves through the arterial system. AIx is the AP expressed as a percentage of central blood pressure, therefore being a combined measure of aortic wave reflection and systemic arterial stiffness [17]. A higher AIx therefore indicates an increase in arterial stiffness based on the higher returning speed and intensity of the reflected wave and is classified as an independent risk factor for CVD [18]. AIx is inversely related to heart rate, with a 10 bpm increase in heart rate accounting for a 4% reduction in AIx [19]. This means that AIx should be normalised for a heart rate of 75 bpm (AIx@HR75) to standardise data and prevent variation [19].

Given the association in other populations, it is feasible to speculate that ME/CFS patients may suffer from impaired arterial function as a result of their illness-related chronic inflammation and oxidation. Spence and colleagues (2008), investigated this concept and found significantly increased ME/CFS mean AIx75 (22.5 ± 1.7) than the control group (13 ± 2.3; *p* = 0.002), indicating increased arterial stiffness by the amplified magnitude and speed of reflected arterial waves [6]. PWV was not directly measured, but an increase in PWV is implied indirectly by the faster return time of the reflected waves [6]. An increased presence of CRP levels was also found in the CFS cohort compared to controls (*p* < 0.01), further implying the correlation of chronic inflammation and artery stiffness in ME/CFS [6]. Recent research from Scherbakov and associates (2020), expanded upon the theory of vascular dysfunction in ME/CFS cohorts and found that peripheral endothelial dysfunction (ED) was also associated with increased ME/CFS symptom severity [20]. The pathogenesis of this mechanism is still not understood but can help provide prognostic information to assess cardiovascular morbidity and mortality in ME/CFS populations [20].

With PEM being the hallmark symptom of ME/CFS, it has been postulated that the presence of exercise/physical activity may have a negative impact on vascular function, resulting in exacerbated symptoms and an increased vascular inflammatory response [21]. One study found a correlation between increased markers of oxidative stress and both high and low-intensity exercise in ME/CFS, suggesting that physical activity can worsen vascular function as opposed to having an anti-oxidative effect seen in healthy individuals [22]. Interestingly, James and colleagues (2016) found that exercise-induced oxidative stress levels were higher in CFS patients who were regularly active (≥6 h per week) prior to CFS onset, as well as in patients who suffered from a severe infection (peritonitis, pneumonia, encephalomyelitis) within 4 months before onset [21]. This research implies that worsened measures of arterial stiffness may be observed in CFS patients while in a state of PEM, yet no study to date has investigated this correlation. Therefore, the aim of the present study was to investigate markers of arterial stiffness before a maximal exercise test and again after a 48 or 72 h recovery period to determine whether onset of PEM may worsen arterial function in ME/CFS in comparison to healthy controls.

## 2. Materials and Methods

This research was undertaken as part of a larger study [23]. The study was performed in the Massey University mobile lab in Tauranga and Cambridge, New Zealand, as well as the exercise labs at Massey University, Wellington, New Zealand. The study was approved by the Centre for Health and Disabilities Ethics Committee Ref:17/NTA/47 and conducted in accordance with the declaration of Helsinki. Written informed consent was obtained from all participants.

### 2.1. Sample Size Calculation

Sample size for the study was calculated for VO_2_ (mL·kg·min^−1^) and work rate (W) [23] as they were significant factors in post-exertional malaise. An analysis of variance (ANOVA) model was used to calculate the sample size where group A (CFS/ME) mean VO_2_ at anaerobic threshold was 22.2 mL·kg·min^−1^, and group B (healthy controls) 28.45 mL·kg·min^−1^, with a standard deviation of 6.19 for groups A and B. For a power output of 80%, a sample size of 16 will be required.

### 2.2. Subjects

A total of 36 participants were recruited for this study. Men and women with ME/CFS (*n* = 20) were paired against age (within 2-year range), gender and activity level matched controls (*n* = 16). From the sample of 32, participants included 10 females (31%) and 22 females (69%), with the higher ratio of females is being normal for ME/CFS populations. However there was only one male in each of the 48 h groups, compared to four males in the 72 h group. Therefore included in the research study were 17 ME/CFS and 16 healthy gender matched controls. Inclusion criteria for ME/CFS participants was determined by whether they met the International Consensus Criteria [24]. Control participants were recruited through email, sourced either through acquaintance of ME/CFS participants, acquaintance of researchers, or from the Massey University staff database. All participants completed a medical/health and pre-exercise questionnaire. Exclusion criteria for both groups included any conditions that contraindicated aerobic exercise [25]. Participants who had a history of fibromyalgia and/or depression were also excluded from the study, due to prominent symptom crossover with ME/CFS.

Before measurements were undertaken, all participants were randomly assorted into one of two groups; a 48 h recovery group (*n* = 10 ME/CFS, *n* = 8 controls) or a 72 h recovery group (*n* = 10 ME/CFS, *n* = 8 controls). Prior to attending testing, participants were advised to avoid food and smoking for 2 h, caffeine for 4 h, and strenuous exercise for 24 h. All participants were also instructed not to engage in exercise between repeated tests.

### 2.3. Assessment of Pulse-Wave Analyisis

Testing commenced after 5 min of resting whilst supine in the mobile lab. Blood pressure was taken twice using a manual sphygmomanometer (Welch Allyn, NZ) and stethoscope (3M Littman, USA). A measure of augmentation index (AIx75), a multifactorial measure of arterial stiffness, was undertaken non-invasively using the validated SphygmoCor waveform analysis system (Cardio X, Australia) [15]. Pressure waveforms were obtained via peripheral tonometry of the radial artery, in which 15 consecutive pulse waves are collected to form an average waveform. From the waveform, the following factors are acquired: (i) augmentation pressure (AP), the added systolic pressure to the pulse wave-form, caused by pressure generated from the speed and magnitude of reflected waves; (ii) AIx, expressed as a percentage of the augmented pressure divided by the pulse pressure; and (iii) AIx75, in which AIx is standardized to a heart rate of 75 bpm to mitigate changes in resting heart rate between participants [19]. AIx measurements were measured in duplicate, at the same time of day for pre and post measurements for each participant to minimise circadian variation. The location and conditions were standardised for both pre and post measures. All measurements were completed by a single experienced operator. AIx measures were examined for accuracy within the software, with the most accurate being selected for the results.

### 2.4. Assessment of Pulse-Wave Velocity

Carotid-radial pulse wave velocity (crPWV) was recorded using the same SphygmoCor device, using a different measurement setting. To determine PWVcr, the pulse wave was recorded via tonometry at both the radial and carotid artery, sequentially. A 3-lead electrocardiogram (ECG) was set up on the participant to simultaneously record ventricular depolarization, indicating the ejection of a pulse wave into the aorta. Transit time was recorded by the operator measuring two surface distances: between the recording site at the carotid artery and the sternal notch (D1) as well as between the recording site of the radial artery and sternal notch (D2). The SphygmoCor calculates the distance travelled by the pulse wave as (D2)-(D1), therefore velocity can be measured as crPWV(m/s) = (D2-D1)/transit time (t). As a rule of thumb, pulse-waves presenting on the device needed to be visually acceptable and have a mean height of 80 mV to be considered a viable measurement. Pulse-wave velocity was measured in triplicate with results reported based on the average across the three measures. Each participants measurements were taken at the same time of day to minimise circadian variation. The location and conditions between pre and post measures were standardised. All measurements were completed by a single experienced operator.

### 2.5. Exercise Protocol

Participants undertook a graded maximal cardiopulmonary exercise cycle test (CPET), outlined by Hodges and colleagues (2020) after baseline measures were recorded. Participants returned for repeated measures either 48 or 72 h after baseline testing, dependent on their allocated group. Resting measures were repeated upon arrival, before completing the second exercise protocol.

### 2.6. Statistical Analysis

Data were summarised and participant characteristics presented in Table 1. All data were assessed for normality with Shapiro and Wilk. Data that were deemed non-normal were treated with non-parametric testing and all other data parametric data. A paired/Wilcoxon test was used to analyse pre and post measures, along with independent/Mann–Whitney U tests where the difference between the two groups were assessed. All data was analysed using IBM SPSS Statistics version 24. Graphs were formulated from the data using Prism (Graphpad).

## 3. Results

Initially, 20 CFS participants were recruited for the study; however, three were unable to complete the entire protocol, one withdrew before the test, two participants had their data corrupted and measurements were unable to be recorded from a further three ME/CFS participants. This also meant that the control subjects matched to these withdrawn participants were excluded from the study. After exclusions, 7 ME/CFS participants and their matched controls remained in the 48 h group, and 4 ME/CFS participants and matched controls in the 72 h group. From the revised cohort of 22, participants included 10 males (45%) and 12 females (55%). There was now only one ME/CFS male in the 48 h group, compared to four ME/CFS males in the 72 h group, meaning that the 72 h group was entirely male.

Table 2 documents the comparison between ME/CFS and healthy controls along with pre and post testing. No significant differences were found across any measure when the entire ME/CFS group was directly compared to the control group at baseline. In conventional resting measures, both ME/CFS groups had the highest mean baseline systolic blood pressure (SBP, 33 mmHg, each). Post-testing demonstrated the 48 h ME/CFS group experienced a mean 5 mmHg decrease in SBP while the 72 h CFS group remained unchanged. The CFS 48 and 72 h groups experienced little to no change in diastolic blood pressure (DBP) in pre-and-post measure (81-81 mmHg and 85–84 mmHg, respectively), while the 48 h control group experienced a mean 3 mmHg decrease in DBP (78–75 mmHg). Surprisingly, the 72 h control group experienced a large mean increase of DBP (76–85 mmHg), but this could be affected by the decreased sample size skewing the means. Central SBP (cSBP) in the 48 and 72 h ME/CFS groups was shown to decrease (119–115 mmHg and 125–117 mmHg, respectively), as well as in the control 48 h group (110–107 mmHg). The 72 h control group however, experienced a large increase in cSBP (112–119 mmHg). None of these changes were significant.

The CFS 48 h group had the highest mean AIx75 at both baseline and at the end of the trial (17.6 and 17.0, respectively) while the CFS 72-h group had a much lower mean baseline and end measure (8.1 and 9.5, respectively). In comparison of baseline values, the ME/CFS 48 and 72 h groups saw little change in AIx75 (0.64% decrease and 1.38% increase, respectively). The 48 h control group however, experienced a significant 7.2% decrease in AIx75 (from 16.6 to 9.42; *p* < 0.005), while the 72 h control group experienced a non-significant increase of 3.5% (6.8 to 10.3). The changes in control baseline measurements are quite extreme, but it should be noted that AIx75 acts on an interval scale, with some measurements reading into negative numbers, skewing results.

Both CFS groups experienced a reduction in crPWV, with a minor change in the CFS 48-h group of 0.56 ms^−1^ (8.86 ms^−1^ to 8.30 ms^−1^) with the CFS-72-h group experiencing a non-significant mean decrease of 1.55 ms^−1^ (8.03 ms^−1^ to 6.48 ms^−1^). The control 48 h group remained almost unchanged (7.52 ms^−1^ to 7.39 ms^−1^) while the control 72 h group experienced a 0.45 ms^−1^ increase (8.55 ms^−1^ to 9 ms^−1^). These changes were not significant.

AIx@75 is a ratio derived from the augmented pressure recorded and the manual pulse pressure recorded, standardized to 75 bpm. Both blood pressure and augmented pressure were recorded twice, with the mean blood pressure being recorded and the augmented pressure being based on the measure with the strongest % accuracy, based on the SphygmoCor reading. Figure 1 presents percentage change in AIx75 at both 48 and 72 h following CPET. The error bars represent standard deviation. Statistical significance was reached only for the control group from baseline to 48 h (*p* < 0.03), and there was no significant difference for the ME/CFS group at 48 h (*p* = 0.7), nor for the control group at 72 h (*p* = 0.2) or ME/CFS at 72 h (*p* = 0.7).

PWV measures were recorded in triplicate for each measure, with the mean recording being taken from measures which met the threshold for recording (pulses having a mean height of 80 mV). Figure 2 represents changes in PWV at 48 and 72 h following CPET. The error bars represent the standard deviation. No statistical significance was reached for the control group from baseline to 48 h (*p* = 0.27), nor for the ME/CFS group at 48 h (*p* = 0.7), or for the control group at 72 h (*p* = 0.3) or ME/CFS at 72 h (*p* = 0.07).

## 4. Discussion

The present study investigated whether the physiological effects of a maximal aerobic exercise test had an impact on baseline markers of arterial stiffness. Therefore, it was hypothesised that worsened measures of AIx75 and crPWV may be observed in ME/CFS compared to controls as a consequence of exercise. This study has shown that ME/CFS patients may not experience systemic arterial vasodilation after aerobic exercise, contrary to what is seen in healthy populations. To the best of our knowledge, this is the first study to assess pre-and-post measures of AIx75 and PWV after a maximal exercise bout in ME/CFS. It has been demonstrated that ME/CFS patients suffer from impaired vascular function, identified from other measures such as the reactive hyperemia scale [20], markers of inflammation and oxidative stress [5,10], heart rate and blood pressure variability [26] as well as AIx75 and PWV [6,27]. Expanded research also suggests that completing physical activity may exacerbate ME/CFS symptoms and provoke an increased oxidative and inflammatory response [21,22].

The main parameters this study aimed to observe were the changes in AIx75 and PWV in the ME/CFS cohorts 48–72 h after a maximal exercise bout. At baseline testing, the 48 h CFS and control group produced comparative AIx75 measurements, as did the 72 h CFS and control group. At 48 h recovery time, the CFS group had experienced very little change in mean AIx75 from baseline, but the control group had experienced a significant reduction in mean AIx75. Healthy populations have shown to experience improvements in AIx75 with only short-term intervals of moderate to high intensity exercise [28], as well as experiencing acute AIx75 reductions as early as 15 min after completing a single bout of exercise [29]. Acute reductions of AIx75 in healthy populations are brought on by vascular vasodilation, a result of relaxation of the smooth muscle in the vascular wall facilitated by exercise-induced nitric oxide, prostaglandins and endothelin production [29]. Research has shown that some ME/CFS cohorts experienced an increase in markers of oxidative stress and vascular inflammation after high-intensity bouts of exercise, as opposed to the anti-oxidative effect seen in healthy populations. This mechanism is thought to be a contributor to the PEM experienced after exercise [21,22]. A similar mechanism may have occurred in our study, which may explain why no improvement was observed in AIx75 in 48- and 72-hour CFS groups, but was in the 48 h control group. Interestingly, The 72 h control group experienced a non-significant increase in mean AIx75 in comparison to their baseline data, indicating a different response to the 48 h control group’s results. We are unsure whether this reflects vascular physiological change from the exercise undertaken in this group, or is a result of having such a small sample size in this specific cohort. Further research with a larger sample size is required to clarify this.

Both of our ME/CFS groups and the 48 h control group experienced minor, non-significant improvements in crPWV. It is well established that continued aerobic exercise training can significantly improve central (aortic) and peripheral (carotid-radial, carotid-femoral), PWV measures in both healthy populations [30], and patients suffering from cardiovascular disease [31]. No current literature exists on observing changes of PWV with exercise in ME/CFS patients, but other studies have iterated that even a single bout of exercise from a stress test can elicit minor improvements (–0.22 ± 0.3 ms^−1^) in central PWV and peripheral systolic/diastolic blood pressure in healthy populations. These effects were reported to last longer than at least an hour, but were not recorded past 60 min [32]. These acute effects are thought to be initiated by the increased blood flow and shear stress against the endothelium during exercise, stimulating endothelial nitric oxide production and producing a vasodilative effect [33]. It may be plausible that this reductive effect lasts up to 48–72 h in CFS patients as well as healthy populations based on our observations. Further research with earlier and more frequent measuring times may provide a clearer pattern to support this hypothesis.

Only two other studies exist that have investigated AIx75 in adult ME/CFS cohorts. The first, from Spence and colleagues (2008), investigated blood markers of inflammation and AIx75 in a ME/CFS cohort (*n* = 41) against a matched control cohort (*n* = 30). They found that their CFS group had significantly greater AIx75 measures than their control counterparts (24.4 ± 1.8 vs. 16.0 ± 2.8, respectively; *p* = 0.017) [6]. Their CFS cohort had a markedly higher mean AIx75 than ours (24.4% vs. 17.64% at baseline, respectively). This large difference in score may be explained by the difference in ME/CFS inclusion criteria. Spence and colleagues (2008), recruited their ME/CFS cohort based on whether they only met the Fukuda classification for CFS [34]. Participants from our study were recruited based on whether they met the International Consensus Criteria (ME-ICC) [24]. The Fukuda criteria for CFS cohorts is considered now to be outdated and flawed. Using Fukuda’s classification, one could meet the criteria for ME/CFS without presenting with the hallmark symptoms of PEM. This flaw can increase the heterogeneity of the patient population and complicate the comparison of patient samples across multiple studies, as seen here [35]. The second study from Witham and associates (2015) is also the only study investigating PWV in an adult ME/CFS cohort [27]. They investigated whether vitamin D3 supplementation would improve markers of vascular function (AIx75 + PWV) and blood markers of oxidative stress in a ME/CFS cohort (*n* = 50). The outcome of their research revealed no change in AIx75 and PWV at baseline and 6 months, but their supplementation cohort had comparatively similar AIx75 and PWV readings to our 48 h ME/CFS group (15% vs. 17.64% and 8.1 ms^−1^ vs. 8.86 ms^−1^, respectively) [27]. The study’s inclusion for ME/CFS participants was based on a diagnosis that fulfilled both the Fukuda criteria [34] and the Canadian Consensus Criteria (CCC) [36]. The CCC and ME-ICC case definitions both require PEM as a pre-requisite symptom for diagnosis and have both been shown to have stronger sensitivity and specificity for diagnosing ME/CFS compared to Fukuda’s original case definition [35]. This means that Witham’s cohort is likely to be a more reliable comparison to our study group, with their participants also suffering from PEM and meeting a more specific diagnosis for ME/CFS. We would likely see a stronger comparison with a larger sample size in our study.

It is important to note that AIx75 and crPWV are not interchangeable measures, despite both being valid predictors of cardiovascular risk [37]. crPWV is considered a local stiffness measure, due to only analysing a single area of vascular flow (between carotid and radial artery). Observed reductions here may only be an indicator of reduced peripheral arterial stiffness. However, AIx75 determines the ratio of augmented pressure to the systemic pulse pressure, providing information of total small vessel and large artery stiffness. This makes AIx75 a systemic measure, reflecting a more global degree of arterial stiffness. For our ME/CFS cohort, this may mean that they may have experienced a slight improvement in peripheral stiffness after a single bout of exercise, but overall large artery stiffness remains impaired or at least unchanged after a single bout of exercise. A larger sample size is needed to determine this pattern in future research.

### Limitations

Several limitations in this study have been identified and will be considered for future research. Firstly, we acknowledge the reduction of size of our sample groups, particularly the 72 h ME/CFS and control groups, due to unforeseen circumstances. Having a larger sample size would improve the clinical significance of our findings and show a stronger data trend. Secondly, we believe that increasing the frequency of our measures and completing at earlier times after the exercise bout (30 min, 1 h, 4 h, 24 h, 48 h, 72 h) would help identify clinically significant acute change in measurements in our ME/CFS cohort, as most other studies investigating acute changes in arterial stiffness after exercise worked within this time-period [32,33]. Finally, we believe that combining our ME/CFS sample sets together instead of splitting them into separate recovery groups will further increase the strength and trends of our findings.

## 5. Conclusions

The current study builds upon the growing evidence that patients with ME/CFS may suffer from impaired vascular function which remains unchanged following exercise. We have shown that ME/CFS patients may experience altered vascular responses following aerobic exercise, contrary to what is seen in healthy populations. This is thought to be a result of the increased presence of oxidative stress and low-grade vascular inflammation, which may be exacerbated by exercise and contribute to the onset of PEM. This information can be further elaborated upon to be potentially used as a diagnostic tool and determine levels of cardiovascular risk in this population. Further research into the effects of exercise on oxidative stress and vascular inflammation is required to verify these claims for ME/CFS.

## Figures and Tables

**Figure 1 ijerph-18-02366-f001:**
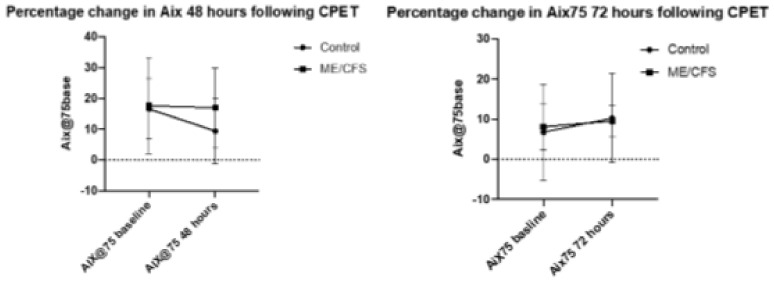
Precentage change in AIx 48 and 72 h following CPET.

**Figure 2 ijerph-18-02366-f002:**
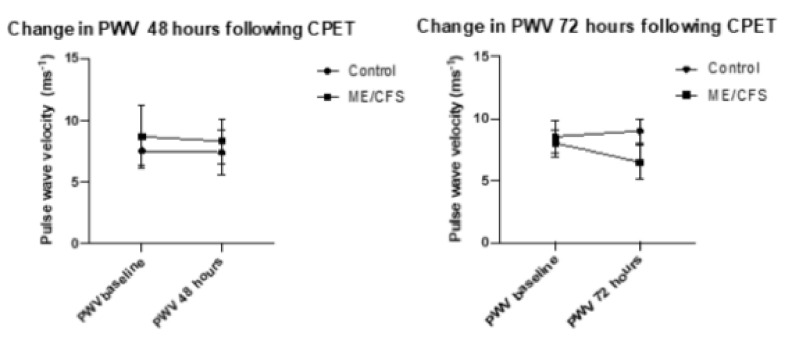
Change in PWV at 48 and 72 h following CPET.

**Table 1 ijerph-18-02366-t001:** Mean (standard deviation (SD)) of participant characteristics.

	ME/CFS48-h *n* = 7	CTRL48-h *n* = 7	*p* Value	ME/CFS72-h *n* = 4	CTRL72-h *n* = 4	*p* Value
**Age** **(years)**	39 +/− 14	45 +/− 11	0.4	56 +/− 6	55 +/− 7	0.8
**Height (cm)**	1.63 +/− 0.07	1.66 +/− 0.08	0.4	1.81 +/− 0.09	1.78 +/− 0.09	0.6
**Weight (kg)**	69.2 +/− 13.1	70.9 +/− 14.4	0.8	85.2 +/− 10.6	87.3 +/− 15.9	0.8
**Body Mass Index (BMI, kg m^2^)**	25.8 +/− 3.2	25.5 +/− 4.2	0.9	25.9 +/− 1.9	27.4 +/− 3.2	0.4

Key: ME/CFS = Myalgic encephalomyelitis/chronic fatigue syndrome, CTRL—controls; ME/CFS48-h = ME/CFS 48-h; CTRL48-h = controls 48-h; ME/CFS72-h = ME/CFS 72-h; CTRL72-h = ME/CFS 72-h; *n* = number.

**Table 2 ijerph-18-02366-t002:** Mean (SD) maximal data of physiological variables from maximal cycle test at baseline and at 48 and 72 h.

		ME/CFS48 h (*n* = 7)	CTRL48 h (*n* = 7)	*p* Value (grp)	ME/CFS72 h (*n* = 4)	CTRL72 h (*n* = 4)	*p* Value (grp)
**SBP** **(mmHg)**	Baseline	132 +/− 9	121 +/− 9	0.97	132 +/− 4	131 +/− 6.8	0.9
	Post	127 +/− 11	122 +/− 14	0.26	133 +/− 5	134 +/− 11	0.9
***p* value (days)**		0.2	0.6		0.6	0.2	
**DBP (mmHg)**	Baseline	81 +/− 10	78 +/− 7	0.62	85 +/− 3	76 +/− 13	0.2
	Post	81 +/− 5	75 +/− 9	0.26	84 +/− 7	85 +/− 4	1.0
***p* value (days)**		0.8	0.5		1.0	0.1	
**cSBP** **(mmHg)**	Baseline	119 +/− 12	110 +/− 12	0.38	125 +/− 10	112 +/− 8	0.2
	Post	115 +/− 11	107 +/− 16	0.32	117 +/− 7	119 +/− 9	0.9
***p* value (days)**		0.2	0.5		0.3	0.1	
**cDBP (mmHg)**	Baseline	82 +/− 9	79 +/− 7	0.62	86 +/− 3	78 +/− 13	0.3
	Post	83 +/− 4	75 +/− 9	0.10	85 +/− 5	86 +/− 5	0.7
***p* value (days)**		1.0	0.6		1.0	0.1	
**AIx75**	Baseline	17.6 +/− 15.5	16.6 +/− 9.8	0.80	8.1 +/− 5.9	6.8 +/− 11.9	0.7
	Post	17.0 +/− 12.9	9.4 +/− 10.6	0.21	9.5 +/− 3.9	10.3 +/− 11.1	0.7
***p* value (days)**		0.7	0.03		0.7	0.2	
**PWV**	Baseline	8.9 +/− 2.5	7.5 +/− 1.2	0.32	8.0 +/− 1.1	8.6 +/− 1.3	0.7
	Post	8.3 +/− 1.8	7.3 +/− 1.8	0.46	6.5 +/− 1.4	9.0 +/− 1.0	0.03
***p* value (days)**		0.27	0.7		0.07	0.3	

SBP—Systolic blood pressure; DBP—diastolic blood pressure; cSBP—central systolic blood pressure; cDBP central diastolic blood pressure; AIx75—augmentation index standardised to a heart rate of 75; PWV—pulse wave velocity.

## Data Availability

The data presented in this study are available on request from the corresponding author.

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
