# Peer review of "Effects of Post-Exertional Malaise on Markers of Arterial Stiffness in Individuals with Myalgic Encephalomyelitis/Chronic Fatigue Syndrome"

_ijerph, 2021, doi:10.3390/ijerph18052366_

Round 1

Reviewer 1 Report

This study investigated exercise impacts on arterial stiffness in patients with myalgic encephalomyelitis/chronic fatigue syndrome (ME/CFS).

The significance and rationale of the study are excellent.

The paper was well written.

However, this study's sample size significantly affected the study's validity, especially with large variances shown in Fig.1.

Furthermore, the post-exertional malaise (PEM) can significantly affect the livings of patients with ME/CFS.

Authors regardlessly randomized the patients into 48-hours and 72-hours recovery groups.

The division reduced the sample size of the study and wasted the efforts of the patients.

A longitudinal study design with three time-points (after exercise, 48 hours, PEM recovery) will be more appropriate.

Reviewer 2 Report

The reviewed manuscript by Bond and colleagues examined the effects of maximal intensity aerobic exercise on vascular function in patients with Myalgic Encephalomyelitis/Chronic Fatigue (ME/CF) Syndrome. Measures of brachial blood pressure, augmentation index normalized to a pulse rate of 75 beats per minute (AIx75), and carotid-radial pulse wave velocity (cr-PWV) were compared between patients and healthy age-matched controls prior to and either 48 or 72 hours following a maximal intensity aerobic exercise stimulus. Though post-exercise measures were similar between patients and controls for most variables, significant reductions in AIx75 at the 48-hour time point were exclusive to the healthy controls. The authors conclude that vascular damage in the ME/CF patients may blunt exercise-induced vasodilation. Overall, I found the manuscript to be well-written and the rationale for the study sound. However, the experimental approach could have been strengthened by a larger sample and comparison of post-exercise vascular responses in all participants at a single time point for greater statistical power.

Major Comments

Abstract

  • The abbreviation for Myalgic Enecphalomyelitis/Chronic Fatigue Syndrome (ME/CFS) should be given after its initial use in the first sentence
  • Because augmentation index can be expressed as either an absolute value, or made relative to a heart rate of 75 beats per minute, I advise using the abbreviation AIx75 or AIx@75 for clarity.

Methods

  • It is stated that 16 ME/CFS patients were recruited, but that 10 were randomized to the 48 hour and 7 were randomized to the 72 hour assessments. These numbers do not make sense, please revise.
  • Post-exercise hypotension is documented to occur within minutes following an exercise bout and persist for up to 2 to 4 hours (PMID: 8225525). Thus, I question the meaningfulness of vascular assessments 48 & 72 hours after exercise. This is even highlighted in the discussion where the authors state that “Healthy populations have shown to experience AIx reductions as early as 15 minutes completing a single bout of exercise” [25].
  • Greater methodological detail regarding PWA and PWV measurements is requested – Were measures obtained in isolation, duplicate, or triplicate? If multiple measures were obtained, how was the reported value decided upon? Were measures obtained by a single or multiple researchers? Was time of day and measurement location/environmental conditions standardized for pre-post measures? Please speak to the level of experience of the individual(s) acquiring these measurements.
  • Differences in physical exertion between the ME/CFS patients and controls may have contributed to post-exercise vascular responses. For example, patients may not have pushed themselves as hard as healthy control subjects. Was this considered? RPE, RER, and/or maximal heart rate data may be used to speak to this point. Also, it would be useful to include CPET testing data (abs and relative vo2max, maximal power, etc.) to compare CRF between groups.
  • A statistical analysis section needs to be added. Due to the small sample size I question the normality of the vascular measures and thus the use of parametric statistical testing.

Results

  • How did the authors decide which control subjects to exclude when attempting to match subject numbers between patients and controls?
  • Age and anthropometric characteristics of the study sample needs to be provided for context
  • A table documenting post-exercise changes in variables such as brachial and central blood pressures is required. Also, these values should be reported as means+/-sd
  •  

Discussion

  • The hypothesis should be included at the end of the introduction section, and the first paragraph of the discussions should highlight the primary finding of the study.
  • To explain the differences between the present study and findings by Spence and colleagues in regards to AIx comparisons between ME/CFS patients and healthy controls the authors point out that criteria used by Spence and colleagues might not be as selective. If this was the case, wouldn’t the more selective criteria used in the present study be anticipated to yield even greater differences between patients and controls?

Reviewer 3 Report

1) Is there an effect of sex on arterial function? Given that the sex distribution is not equal between 48 and 72h groups, sex should be a factor taken into consideration during analyses.

2) There should be a table on participant characteristics including variables such as age, sex, physical activity levels, BMI in the results section. Was BMI/obesity status measured? This could have been a confounding factor for arterial stiffness.  

3) Please include a paragraph on statistical analysis in the methods section.  

4) If oxidative stress/inflammation is the main pathway to impaired arterial stiffness with PEM in ME/CFS patients as discussed widely in the introduction, oxidative stress/inflammatory markers should be measured in this study.  

5) Why was a 48h or 72h recovery period chosen for PEM with exercise? Are there studies to support that arterial function worsens or improves within these time periods?  

6) Suggest to include another figure on BP measurements pre-and post-exercise across controls and ME/CFS groups.

7) Were there controlled measures for the participants (such as exercise restrictions etc.) within the post-exercise period of 48h and 72h, before the post-exercise assessments were done?

8) Definitions of PWV and AIx should be included in the paper and the differences between the two should be mentioned in the introduction. Further explanation on potential mechanisms to why PWV, but not AIx, decreased post-exercise in the ME/CFS group is needed. The authors should also discuss why is there such a trend observed at 72h but not 48h post-exercise?

9) Please include statistical significance in the figures for easy reference to readers. Please also include the meaning of the graphs in the figure caption - whether the data presented refers to the mean/median and what do the error bars represent.

10) In the concluding paragraph, the authors did not measure “systemic arterial vasodilation” with aerobic exercise. Please edit this to reflect the measurement of PWV and AIx which is a surrogate indicator of arterial stiffness/function.

Reviewer 4 Report

The present article aims to examine the impact of maximal-intensity aerobic exercise on vascular function 48 and 72 hours into recovery in patients with Myalgic Encephalomyelitis/Chronic Fatigue Syndrome. The authors found that those with ME/CFS may not experience exercise-induced vasodilation due to chronic vascular damage, which may be a contributor to the onset of post exertional malaise. The article is interesting. I have several minor comments:

First, the authors should state the prevalence of Myalgic Encephalomyelitis/Chronic Fatigue Syndrome.

Second, what kind of cardiovascular abnormalities would patients with Myalgic Encephalomyelitis/Chronic Fatigue Syndrome have? Please be specified.

Third, what is the diagnostic criteria of Myalgic Encephalomyelitis/Chronic Fatigue Syndrome?

Fourth, I suggest the patients should undergo a through ultrasound examination of their musculoskeletal system to rule out other causes of chronic fatigue. Please acknowledge this as a limitation and reference the following two articles: Ultrasound measurements of superficial and deep masticatory muscles in various postures: reliability and influencers. Sci Rep. 2020; Utility of sonoelastography for the evaluation of rotator cuff tendon and pertinent disorders: a systematic review and meta-analysis. Eur Radiol. 2020

Fifth, the methods for statistical analysis should be provided.

Sixth, does the bar in Figure 1 indicate its standard deviation?

Round 2

Reviewer 1 Report

The authors had addressed my comments although the flaws in the study design remains.

Author Response

The authors had addressed my comments although the flaws in the study design remains.

The sample size of the study initially included 20 ME/CFS participants, however three were unable to participate in the entire CPET testing protocol, and one withdrew prior to the first exercise test. From the sample of 32, participants included 10 females (31%) and 22 females (69%), with the higher ratio of females is being normal for ME/CFS populations. However there was only one male in each of the 48-hour groups, compared to four males in the 72-hour group. Therefore included in the research study were 17 ME/CFS and 16 healthy gender matched controls. Baseline fitness levels were similar to each other, as they were for the two control groups. Individuals were randomly assigned to either the 48 or 72- hour study. This was part of another study where the CPET testing results have already been published (Hodges, et al., 2020 The physiological timeline of post exertional malaise in ME/CFS. Translational Sports Medicine, 3 (3): 243-249). 

All-be it there may be small sample size, but this work will help others determine appropriate sample sizes and drop out rates for such studies. At this point it is not possible to change the study design. I can only acknowledge this is the limitations. Please let me know if this would satisfy your requirements. 

Reviewer 2 Report

I would like to thank the authors for their careful consideration of my original comments and commend them on a greatly improved manuscript. I have only one minor concern.

In describing their methodology for the measurement of AIx, the authors state that "AIx measures were examined for accuracy within the software, with the most accurate being selected for the results." Being very familiar with the SphygmoCor software, I can definitely state that the software does not inspect AIx values for accuracy. Thus, I am curious as to how the reported AIx value was selected between the two measures.

Author Response

Thank you for your review. I can confirm that measurements were selected based on an operator index of >90%, with the highest being selected and used in the report. I am happy to re-write the manuscript, but would be pleased to understand if this is acceptable. 

Reviewer 3 Report

  1. Please include the sample size calculation in the paper
  2. The difference in PWV between the control and ME/CFS group at 72h post-exercise but not 48h post-exercise was not explained in the discussion.
  3. While participants were instructed not to engage in strenuous exercise 24h prior to testing, were participants instructed not to exercise within the 48 and 72h period? This could affect the results presented and should be discussed.
  4. The statistical significance are not shown in the figures. Please include it for easy reference to the readers instead of having to refer back to the tables for the p values.

Author Response

Reviewer 3

  1. Please include the sample size calculation in the paper.

The following has now been inserted.

Sample size calculation

Sample size for the study was calculated for VO2 (ml.kg.min-1) and work rate (W)[23] as they were significant factors in post exertional malaise. An anova model was used to calculate the sample size where group A (CFS/ME) mean VO2 at anaerobic threshold was 22.2 ml.kg.min-1, and group B (healthy controls) 28.45 ml.kg.min-1, with a standard deviation of 6.19 for groups A and B. For a power output of 80%, a sample size of 16 will be required.

.

  1. The difference in PWV between the control and ME/CFS group at 72h post-exercise but not 48h post-exercise was not explained in the discussion.

Please could the reviewer be more specific about where this is within the discussion as I am having trouble seeing what is missing and where this needs to go.

  1. While participants were instructed not to engage in strenuous exercise 24h prior to testing, were participants instructed not to exercise within the 48 and 72h period? This could affect the results presented and should be discussed.

The following has now been inserted.

All participants were instructed not to engage in exercise between tests also.

  1. The statistical significance are not shown in the figures. Please include it for easy reference to the readers instead of having to refer back to the tables for the p values.

The following has now been inserted.

AIx75

Statistical significance was reached only for the control group from baseline to 48h (p<0.03), there was no significant difference for the ME/CFS group at 48h (p=0.7), nor for the control group at 72h (p=0.2) or ME/CFS at 72h (p=0.7.) 

PWV

No statistical significance was reached for the control group from baseline to 48h (p=0.27), nor for the ME/CFS group at 48h (p=0.7), or for the control group at 72h (p=0.3) or ME/CFS at 72h (p=0.07). 

Reviewer 4 Report

Good work! The article is acceptable!

Author Response

Many thanks for your review